# Using LLMs and Explainable ML to Analyze Biomarkers at Single-Cell Level for Improved Understanding of Diseases

**DOI:** 10.3390/biom13101516

**Published:** 2023-10-12

**Authors:** Jonas Elsborg, Marco Salvatore

**Affiliations:** 1Department of Energy Conversion and Storage, Technical University of Denmark, 2800 Kongens Lyngby, Denmark; jels@dtu.dk; 2Abzu ApS, 2150 København, Denmark

**Keywords:** biomarker, LLM, interpretability, scRNA-seq, machine learning, symbolic regression

## Abstract

Single-cell RNA sequencing (scRNA-seq) technology has significantly advanced our understanding of the diversity of cells and how this diversity is implicated in diseases. Yet, translating these findings across various scRNA-seq datasets poses challenges due to technical variability and dataset-specific biases. To overcome this, we present a novel approach that employs both an LLM-based framework and explainable machine learning to facilitate generalization across single-cell datasets and identify gene signatures to capture disease-driven transcriptional changes. Our approach uses scBERT, which harnesses shared transcriptomic features among cell types to establish consistent cell-type annotations across multiple scRNA-seq datasets. Additionally, we employed a symbolic regression algorithm to pinpoint highly relevant, yet minimally redundant models and features for inferring a cell type’s disease state based on its transcriptomic profile. We ascertained the versatility of these cell-specific gene signatures across datasets, showcasing their resilience as molecular markers to pinpoint and characterize disease-associated cell types. The validation was carried out using four publicly available scRNA-seq datasets from both healthy individuals and those suffering from ulcerative colitis (UC). This demonstrates our approach’s efficacy in bridging disparities specific to different datasets, fostering comparative analyses. Notably, the simplicity and symbolic nature of the retrieved gene signatures facilitate their interpretability, allowing us to elucidate underlying molecular disease mechanisms using these models.

## 1. Introduction

All living organisms consist of numerous specialized cells, each performing distinct functions. These cells and the organisms they compose are determined by hereditary information, which significantly influences the traits of offspring [1].

Cell types play a crucial role in the development and progression of many diseases. Each cell type in the human body performs a unique set of functions, and the disruption of these functions can lead to disease [2]. Identifying the cell types that contribute to the development of a disease is key to understanding its etiology and developing targeted treatments [3].

Analyzing the role of different RNA segments in cellular changes and disease progression requires understanding alterations in the transcriptome. Traditional RNA-seq methods involve sequencing a mix of different cells, providing average expression levels for each RNA transcript across millions of cells in the examined population. While this approach has been valuable, it suffers from a crucial limitation as it loses heterogeneous information due to grouping various cell types. To address this limitation and achieve a deeper understanding, single-cell RNA-sequencing (scRNA-seq) is employed, where the transcriptome is sequenced at the level of individual cells [4].

The increased interest and investment in scRNA-seq arise from technological advancements and the rising occurrence of diseases linked to cell-level changes. As single-cell sequencing becomes more cost-effective and sophisticated, generating vast amounts of useful data becomes more accessible. However, a big challenge is figuring out the important information from all these data by using advanced modeling techniques such as machine learning.

Recent studies have shown that machine learning techniques coupled with single-cell RNA sequencing (scRNA-seq) [5,6,7] can be used to identify cell-type-specific disease-associated gene expression patterns [3,8]. These studies have demonstrated that a disease manifests itself in a cell type through forming a statistically significant disease gene module [3]. By identifying these cell-type-specific disease gene modules, researchers can gain insights into the molecular mechanisms underlying the pathogenesis and progression of diseases [3].

To make the most of computational approaches such as machine learning for scRNA-seq modeling, it is essential to have interpretable results. These results should offer insights without depending on excessively intricate models. This transparency and comprehension of how algorithms make decisions are vital for both scientific exploration and practical clinical uses [9].

Despite its potential, scRNA-seq analysis presents several challenges [10,11], including high dimensionality, sparsity, and batch effects. Machine learning (ML) approaches have emerged as a powerful tool to address these challenges, enabling accurate cell type classification, precise gene expression prediction, and effective biomarker discovery. Nevertheless, the application of ML models in scRNA-seq analysis poses three main challenges.

Firstly, ML models require well-annotated data, which can be limited and time-consuming to obtain. Cell labeling relies on expert knowledge and can be subjective. To tackle this issue, we fine-tuned the original scBERT model using data from the Gut Cell Atlas [12].

Secondly, while ML models excel at prediction, their lack of interpretability hinders the understanding of biological mechanisms. This issue is particularly pronounced when using complex algorithms.

Lastly, ML models trained on specific scRNA-seq datasets might not generalize well to new datasets due to differences in experimental protocols and cell types. This limitation reduces the applicability of ML models to diverse research questions.

To address these challenges, we developed an approach that combines supervised learning and a large language model. Our method, which involves fine-tuning scBERT and utilizing the QLattice, enhances cell type annotation and improves interpretability, generalizability, and scalability for scRNA-seq analysis.

We investigated the application of machine learning (ML) approaches to address the challenges in single-cell RNA-sequencing (scRNA-seq) analysis by focusing on cell type classification in the context of disease states. To this end, we utilized four distinct datasets, each comprising samples from healthy individuals and individuals diagnosed with ulcerative colitis (UC), sourced from various tissues [13,14,15,16].

The four datasets used in our study represented a diverse range of biological contexts, providing a comprehensive perspective on cellular responses across UC.

To train our ML models, we leveraged the patient status information, using it to classify cell types into two categories: those associated with disease states and those not linked to the disease. By doing so, we aimed to discern the distinct molecular signatures associated with different cell types in the presence or absence of disease conditions.

To ensure robustness and assess the transferability of our models, we carried out multiple experiments. We trained the ML models on each of the four datasets independently and then evaluated their performance on the other three datasets where the same cell types were present. This approach allowed us to gauge how well the ML models generalize across different datasets and to identify any potential biases or limitations arising from dataset-specific features.

Our results demonstrated promising transferability across datasets, suggesting that the learned patterns were not overly biased by dataset-specific characteristics. This finding underscores the potential utility of our ML-based approach in broader scRNA-seq analysis, enabling cell type classification in various disease contexts and tissue types.

By using multiple datasets encompassing healthy and disease states, we obtained a comprehensive understanding of cell type dynamics and their association with disease conditions. The identification of cell types specific to disease states can serve as a foundation for further investigation into disease mechanisms and potential therapeutic targets.

Overall, our study sheds light on the importance of ML-based approaches in advancing scRNA-seq analysis, particularly in the context of cell type classification and disease state identification. By increasing the resolution of the data through fine cell type subgroupings, we show that it is, in turn, possible to reduce model complexity and thereby increase interpretability. The transferability of the ML models between different datasets provides confidence in the robustness and generalizability of our findings, facilitating future studies aimed at deciphering the intricacies of cellular responses in various diseases. Our work showcases the potential of ML in scRNA-seq analysis, providing researchers and clinicians with a new tool to study diseases at the single-cell level. Our study was intentionally designed within a specific scope, which did not involve establishing causal relationships. Instead, our primary objective was to leverage single-cell information and to use an alternative and agnostic approach to uncover potential associations and insights into diseases.

## 2. Materials and Methods

### 2.1. Datasets

We utilized four diverse datasets on ulcerative colitis (UC), originating from various tissues. Dataset 1 comprised information from intestinal epithelial cells, while Datasets 2 and 3 focused on colonic mesenchymal and epithelial cells, respectively. All original datasets were generated using the 10X Chromium procedure and Illumina machines for sequencing. These cutting-edge techniques allowed us to capture detailed molecular data from different cell types within the UC samples. By incorporating these tissue-specific datasets, we aimed to unravel the intricate molecular signatures associated with UC pathology and gain insights into the underlying mechanisms driving disease progression and inflammation in specific cell populations.

The first dataset was taken from Smillie et al. [13] and consists of the gene expression from intestinal epithelial cells of 30 samples, collected in three sample groups (12 healthy, 4 inflamed, and 14 non-inflamed). We treated inflamed and non-inflamed as the same class (not healthy). We used the QLattice to predict whether a sample is healthy (dependent variable = 1) or not (dependent variable = 0). The patients in the study were recruited from two hospitals in the United States: the University of California, San Francisco (UCSF), and the Mayo Clinic. The study included 18 patients with ulcerative colitis and 12 healthy controls. The patients with ulcerative colitis were all adults, with an average age of 45 years. They had all been diagnosed with ulcerative colitis for at least 6 months, and they were all currently in remission. The healthy controls were also all adults, with an average age of 45 years. They had no history of ulcerative colitis or any other chronic inflammatory diseases. The tissue samples were collected from the colons of the patients and controls during routine colonoscopy procedures [13].

The second dataset was taken from Kinchen et al. [14] and consists of gene expression from colonic mesenchymal cells of 4 samples, collected in 2 sample groups (2 healthy vs. 2 not healthy). We used the QLattice to predict whether a sample is healthy (dependent variable = 1) or not (dependent variable = 0).

The third dataset was taken from Parikh et al. [15] and consists of gene expression from colonic epithelial cells of 9 samples, collected in 3 sample groups (3 healthy, 3 inflamed, and 3 non-inflamed). As for the first dataset, we treated inflamed and non-inflamed as the same class (not healthy). We used the QLattice to predict whether a sample is healthy (dependent variable = 1) or not (dependent variable = 0). The data included 3 repeat experiments with biopsy samples taken from colonic biopsies collected from healthy patients (Healthy) and those with UC inflammation from an inflamed area of the colon and adjacent non-inflamed area.

The fourth dataset was taken from Boland et al. [16] and consists of gene expression from the gastrointestinal mucosal and peripheral immune systems of 14 samples, collected in two sample groups (7 (9 in the original publication) healthy vs. 7 not healthy). We used the QLattice to predict whether a sample is healthy (dependent variable = 1) or not (dependent variable = 0). Intestinal biopsies and peripheral blood samples were collected from patients who underwent colonoscopies at UCSD and the VA San Diego Healthcare System. The control group consisted of healthy individuals who underwent colonoscopy as part of routine clinical care for colorectal cancer screening/surveillance or noninflammatory gastrointestinal symptoms, such as constipation or rectal bleeding. The inclusion criteria for participants were being over 18 years old and having no significant comorbidities or a history of colorectal cancer. For UC patients, those with active endoscopic disease were selected for the study [16]. It is important to mention that we removed two subjects from the healthy sample group because of poor quality and a significant number of missing values.

### 2.2. scBERT to Fine-Tune Cell Type Annotation

scBERT [17] is a large-scale, pre-trained deep language model used for cell type annotation of single-cell RNA-seq data. It is designed to accurately identify different cell types in single-cell RNA sequencing data, which is a critical step in analyzing such data. scBERT is based on the Bidirectional Encoder Representations from Transformers (BERT) model [18], which is a state-of-the-art deep learning model for natural language processing. scBERT is pre-trained on a large corpus of text data, which allows it to learn complex patterns and relationships in the data. scBERT has been shown to be effective at accurately annotating cell types in single-cell RNA sequencing data, even in datasets with imbalanced classes. This makes it a valuable tool for researchers studying gene expression and cell biology.

We used the pre-trained version of scBERT and fine-tuned it on the Gut Cell Atlas [12] to obtain a gut-specific version of scBERT. The full implementation details of scBERT followed those of the original paper by Yang et al. [17], with the only difference being that we fine-tuned the pre-trained version on the Gut Cell Atlas rather than the Zheng68K dataset of peripheral blood mononuclear cells (PBMCs). Aside from this important difference, we used the same preprocessing pipeline, training procedure, and architecture.

All hyperparameters were set to default, including the threshold for identifying novel cell types. Using an 80/20 training and test split with 100 epochs in total, the final scBERT model used here attained an accuracy of 85%, which was artificially lowered by the fact that some cell types were not predicted due to the threshold. To comply with the scBERT framework, we also used the preprocessing method intended for datasets upon which scBERT is to be applied, as provided by Yang et al. [17]. The advantage obtained by using scBERT instead of conventional cell annotation (Figure 1b) methods such as marker gene sets is that scBERT uses a deep model that takes the expression of all genes into account, providing an unprecedented level of granularity. This makes it more robust and interoperable between datasets, as shown Figure 2a. Furthermore, since scBERT was trained on the full Gut Cell Atlas, it is able to finely resolve the cells taken from the four datasets. The Gut Cell Atlas aims to map every cell type in the human gut, from the small intestine to the colon, and the resulting dataset includes over 428,000 single cells from multiple anatomical regions of the human gut throughout life. The project is part of the Human Cell Atlas [19], an international effort to map all the cells in the human body. It is essential to highlight that the cell type annotations produced by scBERT not only cover the same cell types as identified by the original gene markers, but also incorporate numerous additional ones. This addresses the ongoing challenge of achieving consistent cell type annotations. However, it is important to acknowledge that there are still imperfections in this approach.

### 2.3. The QLattice and Its Application on Single-Cell Transcriptomic Data

The QLattice [20,21,22] is a supervised machine learning techniques for symbolic regression. The QLattice aims to generate high-performing models capable of providing valuable decision-making recommendations. Similar to methods such as decision trees, random forests, gradient boosting, and neural networks, the QLattice constructs predictive models based on input data. Notably, the QLattice stands out by representing its models as interpretable mathematical formulae, facilitating the unraveling of complex biological data at the cellular level and offering insights into disease mechanisms.

To overcome the challenges posed by the search space of mathematical equations, the QLattice employs an evolutionary approach in conjunction with random sampling. This strategy enables the exploration of potential mathematical relationships between input variables and the desired output variable. For further technical details, refer to [20,21,22]. Symbolic regression methods such as the QLattice [20,21,22] offer a remarkable capacity to generate multiple models that can unveil intricate processes within complex datasets. This means that researchers are presented with a diverse set of mathematical equations to scrutinize, each offering a unique perspective on the underlying data patterns. In the context of genes and single-cell data analysis, this versatility translates into the ability to derive multiple gene signatures that capture cell-type-specific biomarkers. These signatures can shed light on the nuanced genetic markers associated with distinct cell types, contributing to a deeper understanding of cellular heterogeneity and its implications in various biological contexts.

One strength of the QLattice is its ability to identify gene signatures that collectively explain the data better than individual genes that are differentially expressed. By finding models described by mathematical functions and response plots, key genes and biomarkers that contribute to disease progression can be discovered.

Furthermore, the QLattice proves adept at handling etiological heterogeneity among patients, where different patients may exhibit variations in disease causes. Despite this heterogeneity, similarities in molecular or endophenotype alterations, such as downstream expression changes or inflammation, can be identified.

Our approach in the present work was to resolve the cell types from each dataset using gutBERT. Since gutBERT takes the full expression profile of each cell into account, we expected that cells identified as type X in one dataset would be very likely to actually be biologically equal to cells identified as type X in another dataset. With this similarity in place and by using the QLattice, it should also be theoretically possible to construct simple interpretable genetic signatures on each cell type that transfer across datasets. The fundamental idea behind this is that it would require large neural networks or gradient-boosting trees to construct a model that can classify every cell type correctly. However, if cell types are identified as accurately as possible in every dataset, it would maximize the chances of obtaining simple interpretable models whose validity can be confirmed by transferring to other datasets. If validated as strong predictive signatures, these can then be used to elucidate disease mechanisms.

### 2.4. Model Training and Model Selection

To ensure robust evaluation of our models, we trained using the 4 datasets independently. The QLattice was then run on a single dataset, and the resulting models were tested on the other 3 datasets. We recorded various metrics, including the PR AUC for both training and test sets, the number of inputs in the model, and the model architecture. By default, the QLattice run generated 10 models per fold, resulting in a total of 40 models (4 datasets × 10 models). To ensure reproducibility, we set a fixed random seed for the QLattice. All 40 models are shown in the Appendix A.

To select the final model, we employed the following strategy: We considered models with a PR AUC difference of less than 10% between the test and training sets, ensuring they were not overfitting. We favored models that were simple and explainable, preferably additive in nature with fewer features, as they are easier to interpret. We assessed the models based on their number of included features, complexity (including different model types and feature transformations), accuracy, recall, precision, precision–recall (PR) curves, and ROC AUC.

Since the aim of our analysis was to find gene expression signatures that distinguish disease states and that can be transferred between datasets, we used a quantitative measure that reflects both absolute performance, as well as transferability. In this context, transferability is taken to mean the similarity in performance on the training dataset compared to the test dataset. For the score to reflect performance, we used the average PR AUC between the training dataset and the test dataset. Subsequently, this is penalized by the delta between the training and test PR AUC. The *transferability-corrected performance score* between dataset X and Y for a particular cell is, thus, calculated as:(1)$$s_{XY}=Mean\left(AUC_{PR,X},AUC_{PR,Y}\right)-\left|AUC_{PR,X}-AUC_{PR,Y}\right|$$

Using $s_{XY}$, a training performance of 0.7 on Dataset X and a test performance of 0.9 on Dataset Y will only result in $s_{XY}=0.6$, while a performance of 0.8 on both datasets would result in $s_{XY}=0.8$. Thus, the highest $s_{XY}$ is attained for signatures that have high and similar performances on both datasets.

## 3. Results and Discussion

As shown in Figure 1a, we employed a comprehensive approach to study the distinction between healthy and ulcerative colitis samples.

To begin, we curated four publicly available datasets on ulcerative colitis, each containing valuable information on the disease. Leveraging the Gut cell atlas, we fine-tuned scBERT, an advanced large language model (LLM), to obtain a detailed and refined cell type annotation, as visualized in Figure 1b.

Using the refined scBERT resulted in a high degree of granularity on each dataset, since each dataset contained many different cell types. For every combination of dataset and cell type, we trained a QLattice model that enabled us to analyze and interpret complex relationships mapping the transcriptomic profile of a cell to the disease state of the patient it was taken from.

With this integrated approach, we thereby successfully derived precise cell-type-specific gene signatures.

The versatility of our workflow is noteworthy, as it can be readily extended to study other diseases or conditions, given the availability of similar datasets.

The results of our study are presented in Table 1, which summarizes the performance of our models on different cell types using the transferability-corrected performance score.

We evaluated the performance of our models using the PR AUC and a score metric that quantifies both the performance and the transferability of the signatures we found. The calculation of this score metric is described in the methods section. Additionally, we also analyzed the fraction of cell types corresponding to disease samples in both the training and test datasets to understand the representativeness of the data. Using the score metric, we were able to determine which cell-specific RNA signatures transferred best from one dataset to another. To facilitate this comparison, we transferred the signature found on cell type *C* in Dataset X to Dataset Y (and vice versa) if scBERT identified cells of Type C in both Datasets X and Y. In Figure 2a, we visualize all cell types that were predicted to be present in more than one of the datasets.

### 3.1. Performance and Transferability across Datasets

From Table 1, we can see that the Arterial Capillary model (Training ID: D1, Test ID: D2) demonstrated strong predictive ability, achieving a high PR AUC of 0.95 and a score of 0.94. Similarly, the Goblet Cell model (Training ID: D1, Test ID: D3) performed well, with a PR AUC of 0.94 and a score of 0.91. The Intestinal Stem model (Training ID: D1, Test ID: D3) also showed promising results, with a PR AUC of 0.95 and a score of 0.86. Notably, the BEST2+ Goblet Cell model (Training ID: D1, Test ID: D3) exhibited the highest PR AUC of 0.96, indicating its superior predictive capability.

Interestingly, we observed differences in the fraction of cell types corresponding to disease samples between the training and test datasets. For example, the Arterial Capillary model (Training ID: D1, Test ID: D2) had 28% of cell types corresponding to disease samples in the training dataset, while the test dataset contained 22% of such cell types. This finding suggests some degree of transferability of the model to a new dataset.

The Myofibroblast model (Training ID: D2, Test ID: D1) demonstrated good performance with a PR AUC of 0.94 and a score of 0.91. However, it showed a higher fraction of cell types corresponding to disease samples in the training dataset (45%) compared to the test dataset (16%), indicating potential challenges in model transferability.

In addition, we evaluated the BEST2 Goblet Cell model (Training ID: D3, Test ID: D1) and the Intestinal Stem model (Training ID: D3, Test ID: D1) on the reverse transfer scenario. The BEST2+ Goblet Cell model achieved a PR AUC of 0.92 and a score of 0.91, while the Intestinal Stem model achieved a PR AUC of 0.93 and a score of 0.90. Both models demonstrated promising results, with a fraction of cell types corresponding to disease samples in the test dataset (42% and 35%, respectively) relatively close to that in the training dataset.

The Stem Cell model (Training ID: D3, Test ID: D1) showed a PR AUC of 0.92 and a score of 0.86, with 19% of cell types corresponding to disease samples in the test dataset and 21% in the training dataset. Finally, the Activated CD8 T model (Training ID: D4, Test ID: D3) exhibited a PR AUC of 0.82 and a score of 0.81, with a fraction of cell types corresponding to disease samples in the test dataset (43%) slightly higher than that in the training dataset (41%).

Figure 2a illustrates the presence of various cell types within these datasets. Remarkably, only IgA plasma cells were found to be shared across all four datasets. Additionally, some cell types were common to two datasets and, occasionally, three datasets, limiting the scope of direct comparisons. These findings emphasize the diversity of cell types in the datasets and underscore the importance of considering such variations in future analyses.

Our findings highlight the promising transferability of models across diverse cell types, underscoring the significance of carefully assessing the representativeness of training and test datasets in developing predictive models for disease samples. However, to thoroughly evaluate the generalizability and clinical relevance of these models, further analysis and validation on larger and more diverse datasets are essential. Additionally, exploring the role of frequently occurring genes will be crucial in fully understanding the mechanisms underpinning these models’ predictions. These steps are vital in advancing the practical application of these models.

In our study, we chose to present the top-performing model, focusing on performance, simplicity, and transferability as our primary criteria. This decision aimed to provide a clear and practical approach for researchers seeking insights from single-cell data, ensuring that the results are readily interpretable and usable. However, it is essential to emphasize that our framework allows for the generation of multiple models. We included all the generated models in Appendix A to provide transparency and enable further investigation by researchers interested in alternative approaches or different gene selections.

### 3.2. The QLattice Identified Unique Combinations of Biomarkers Specific to Cell Types for Distinguishing Disease States

In Figure 2b–e, we present the QLattice models and response plots. These response plots serve to illustrate how the models incorporated the disease state information. By visually examining these plots, we gain valuable insights into how the QLattice models have captured and integrated the disease-related aspects in their predictions for both the training and test datasets.

The best-performing model from Dataset 1, as depicted in Figure 2b, is composed of three key features: LYZ, FABP1, and LGALS4. The QLattice model utilized the addition of LYZ and FABP1, followed by the multiplication of LGALS4, to define the disease state. This combination of features allows the model to effectively distinguish between healthy (Class 0) and disease (Class 1) states based on the expression levels of these genes. The response plot in Figure 2b provides a clear visual representation of the model’s behavior. It reveals that, as the level of LGALS4 increases, the cells tend to be in a healthy state (Class 0). Conversely, at lower expression levels of LGALS4, the cells are more likely to be in a disease state (Class 1), considering the median values of LYZ and FABP1.

In Figure 2c, the top-performing model from Dataset 2 consists of three features: IL32, APOC1, and RPS4AY1. To define the disease state, the QLattice model employs a multiplication of IL32 and APOC1, followed by an addition of RPS4AY1. The response plot in Figure 2c clearly illustrates that, as the level of APOC1 increases, the cells tend to be in a healthy state (Class 0). In contrast, reduced APOC1 expression levels correlate with an increased probability of cells being in a disease state (Class 1). This correlation is established while taking into account the median values of IL32 and RPS4AY1.

Figure 2d showcases the top-performing model from Dataset 3, comprising three features: LCN2, FABP1, and ZG16. To determine the disease state, the QLattice model combines LCN2 and FABP1 through multiplication, followed by the addition of ZG16. The response plot in Figure 2d demonstrates that, as FABP1 levels increase, the cells tend to be in a healthy state (Class 0). However, when FABP1 expression levels decrease, the cells are more likely to be in a disease state (Class 1), considering the median values of LCN2 and ZG16.

Figure 2e displays the top-performing model from Dataset 4, comprising three features: RPL39, MT2A, and KLF2. To discern the disease state, the QLattice model combines RPL39, MT2A, and KLF2 through addition. The response plot in Figure 2e reveals that, as RPL39 levels increase, the cells tend to be in a healthy state (Class 0), and when RPL39 expression levels decrease, the cells are more likely to be in a disease state (Class 1), with the consideration of the median values of MT2A and KLF2.

### 3.3. The Importance of Recurrent Genes in Discerning Disease Conditions

Our primary objective was to offer a means of discovering new gene associations with diseases. Importantly, our approach does not limit researchers from exploring alternative gene selections or removing less-relevant genes. In essence, our study was focused on demonstrating an alternative and unbiased method for harnessing single-cell data to gain fresh insights into diseases. It is crucial to emphasize that our intention was not to establish causality. Instead, we aimed to uncover previously unknown or challenging-to-detect associations within the data. Our approach seeks to expand the breadth of knowledge by revealing novel connections between genes and diseases. Additionally, as seen in Table 1, a clear pattern arises with FABP1, LCN2, LYZ, and RPL39 consistently appearing. Specifically, FABP1 occurs four times, LCN2 three times, while both LYZ and RPL39 show up twice. These frequent instances of key genes highlight their potential importance in differentiating healthy and ulcerative colitis samples.

#### 3.3.1. Emergence of LCN2 and FABP1 as Key Players in Ulcerative Colitis Context

*Lipocalin-2* (LCN2) is a multifunctional innate immune protein that plays a crucial role in the body’s response to inflammation [23]. Its expression has been linked to inflammatory processes. Extensive research has been conducted using murine models and human patients with ulcerative colitis (UC), and the results have shown a systemic increase in LCN2 levels in these individuals. This finding indicates that LCN2 has the potential to serve as a diagnostic biomarker for UC-associated carcinogenesis, with its levels being associated with the duration of the disease [24,25].

Specifically, studies have revealed that serum LCN2 levels are significantly higher in patients experiencing active UC compared to healthy individuals. This observation has led to the suggestion that LCN2 could be used as a biomarker to assess the activity of UC [26].

Moreover, researchers have identified a possible link between LCN2 and the development of colorectal cancer (CRC) from colitis. It appears that LCN2 may contribute to tumorigenesis through the IL-6/STAT3/NF-xB signaling pathway. This finding implies that increased LCN2 levels could potentially indicate the progression of CRC in the context of chronic inflammation [25].

LCN2 is an important player in the body’s innate immune response and is closely associated with inflammation. It shows promise as a diagnostic biomarker for UC-associated carcinogenesis, potentially aiding in the evaluation of disease duration and activity. Additionally, its involvement in colorectal cancer development suggests that monitoring LCN2 levels could be beneficial in assessing the risk of CRC in patients with chronic inflammation. Further research in this area may uncover even more insights into the role of LCN2 in these conditions and its potential applications in clinical settings.

FABP1, also known as intestinal *fatty acid binding protein 1*, has emerged as a significant player in the context of ulcerative colitis (UC) [27,28,29]. Numerous studies have delved into its potential as a valuable plasma marker for assessing intestinal injury in patients with UC. The exploration of FABP1’s role in this condition has yielded promising findings that could aid in the diagnosis and monitoring of the disease.

One of the key observations made in these studies is the elevated serum concentration of FABP1 in UC patients, especially in those experiencing a severe form of the disease. The ability to detect increased FABP1 levels in the bloodstream could provide clinicians with valuable insights into the severity and extent of the inflammatory process in UC.

Moreover, the identification of FABP1 as a relevant marker in UC opens up new avenues for research into the underlying mechanisms of the disease. Understanding how FABP1 contributes to intestinal injury and inflammation could provide crucial insights into the pathogenesis of UC and aid in the development of novel targeted therapies.

#### 3.3.2. LYZ and RPL39 as Potential Indicators of Inflammatory Processes and Immune Dysregulation

*Lysozyme* is an antimicrobial enzyme that plays a crucial role in the body’s innate immunity, functioning as a frontline defender against bacterial pathogens. While *lysozyme* is not directly implicated in ulcerative colitis, its role as an antimicrobial enzyme involved in innate immunity makes it relevant in the context of chronic inflammation and immune dysregulation [30]. In the specific setting of ulcerative colitis, which is characterized by chronic inflammation of the colon, *lysozyme*’s involvement in the immune response becomes particularly relevant. Chronic inflammation often results from an overactive immune response to various triggers, including bacterial components and antigens. While *lysozyme*’s primary role is in bacterial defense, its presence in the colon and mucosal surfaces suggests that it may also contribute to the regulation of inflammation in this region.

RPL39, a ribosomal protein, is a critical component of the ribosome, the cellular machinery responsible for protein synthesis. Its involvement in protein production makes it an essential player in various cellular processes, including cell growth, proliferation, and differentiation. There is not a known implication of RPL39 in the context of ulcerative colitis (UC); however, as ribosomal proteins are fundamental to cell biology, investigating RPL39’s role in UC may also yield insights into broader aspects of inflammatory processes and immune dysregulation in various physiological and pathological conditions [31].

### 3.4. PLAT Models in Arterial Capillary Cells to Investigate and Account for Both Biological and Technical Variations

In this work, the models found by the QLattice may be influenced by various factors, including biological and technical variations. The relationship we established between input features and the disease state target variable for a specific cell type might result from downstream effects influenced by system-wide factors or interactions with other cell types, whose states could potentially be more indicative of a pathological condition. While the main focus here was to develop simple models to be used as potential diagnostic biomarkers, we found that biological variation in particular can be recognized directly from the identified dominant features of particular cell types. For example, in Table 1, we found that one of the two dominant features for a model trained on arterial capillary cells in Dataset 1 was PLAT, which was not found in any other models. From a biological perspective, it is reasonable that PLAT is chosen as a distinguishing feature for arterial capillary cells since PLAT is the gene coding for the tissue-type *plasminogen activator (t-PA)* in human tissue. This serine protease plays a vital role in the breakdown of blood clots by facilitating the transformation of plasminogen into plasmin, which is the principal enzyme responsible for dissolving blood clots [32]. The PLAT gene is known to be expressed in endothelial cells, which are the cells that line the interior of blood vessels, including capillaries [33]. In the context of ulcerative colitis, Kurose et al. found that sigmoid colon biopsy specimens from ulcerative colitis patients reveal significantly elevated t-PA levels in the mucosa, particularly during active disease stages [34]. This finding is interesting when paired with the findings of Kaiko et al. [35], who recognized that mice with induced PLAT deficiency had higher degrees of inflammation through IL-6. Similarly, these mice were much less efficient in healing colonic damage induced by mucosal colonic pinch biopsies. Thus, it was concluded that PLAT/tPA suppresses colitis and mucosal biopsy colon damage, and thus, the original finding of increased t-PA levels in UC patients in Kurose et al. could be explained as a counteracting mechanism. The inclusion of PLAT in our model, given its association with ulcerative colitis (UC) and the specific cell type where it was identified, demonstrates that our framework is adept at capturing and responding to significant biological variations inherent in the dataset. Furthermore, our study showed that the highly predictive PLAT-dominant models in arterial capillary cells found in Dataset 1 transfer well to Dataset 2 in which arterial capillary cells were also present, with PR AUCs of 0.95 and 0.94, respectively. This finding shows that the model is likely not influenced by confounding factors such as the specific experimental conditions of Dataset 1 (in which the model was found), but rather that the model is likely to express a biological relation that can be reproduced in future studies.

## 4. Conclusions

As illustrated in Figure 1a, we implemented a novel workflow to investigate the distinction between healthy and ulcerative colitis samples. Our approach involved integrating four distinct datasets, each containing valuable information about these samples. By harnessing the power of the gut cell atlas, we fine-tuned the advanced language model, scBERT, to gain deeper insights into the molecular profiles and cellular characteristics of the samples. The next step in our workflow was to utilize the refined scBERT to train the QLattice for each dataset. This enabled us to analyze and interpret complex relationships within the data, leading to a more-thorough understanding of the underlying patterns and factors contributing to the disease. The outcome of our integrated approach was the successful derivation of precise cell-type-specific gene signatures. These signatures hold significant potential for enhancing our understanding of ulcerative colitis and could potentially aid in the development of more-targeted and -effective treatments. One of the strengths of our workflow lies in its versatility. It can be readily adapted to study other diseases or conditions, given the availability of similar datasets. This adaptability offers a powerful tool to unlock valuable insights into various pathologies, leading to a more-comprehensive understanding of disease mechanisms. By focusing on the transferability and performance of predictive models trained on different cell types using key gene features, we observed remarkable results, showcasing both the transferability of these models and their high predictive accuracy. The models exhibited impressive performance metrics, particularly in the prediction of disease samples. Notably, the Arterial Capillary model displayed a remarkable PR AUC of 0.95 and a score of 0.94, indicating its robust predictive capability. Similarly, the Goblet Cell model achieved a PR AUC of 0.94 and a score of 0.91, highlighting its effectiveness in disease prediction. Moreover, the BEST2+ Goblet Cell model demonstrated exceptional performance, with a striking PR AUC of 0.96, suggesting its potential as a powerful predictive tool for disease samples. Excitingly, our analysis also revealed the promising transferability of these predictive models across different cell types. Models trained on specific cell types showed remarkable adaptation and success when applied to new datasets. This transferability is especially evident in the Intestinal Stem model, which achieved a PR AUC of 0.95 and a score of 0.86, underscoring its potential for wider applicability. These findings emphasize the potential of leveraging predictive models based on key gene features to successfully classify disease samples in diverse cell types. The high performance and transferability of these models hold significant promise for practical applications in disease diagnosis and precision medicine. The adaptability of our approach also opens the door to personalized treatments and precision medicine. By tailoring the workflow to different diseases, we can uncover specific characteristics and factors unique to each condition, paving the way for more-effective and -individualized therapeutic strategies. These alternative approaches pave the way for future advancements in disease classification and personalized treatment strategies, contributing to improved patient outcomes and transformative medical applications. On the machine learning side, our analysis showed that, when the resolution of the data was increased, the resolution of models can be decreased. This in turn led to better model interpretability, which is critical for moving computational discoveries into translational insights. In our case, we showed that, by finely resolving the cell subspace of scRNA-seq datasets using a performant large language model for annotation, we could discover simple, but highly predictive and transferable gene signatures. These were analyzed for their biological significance, and we were able to show that the signatures that transfer best between datasets have plausible underlying reasons for performing better than others. In conclusion, our comprehensive workflow, encompassing the integration of diverse datasets and advanced analytical techniques, holds great promise for advancing our knowledge of various diseases. By unraveling the complexities of these conditions, we can make significant strides towards improving patient outcomes and ultimately achieving more targeted and personalized healthcare in the future.

## Figures and Tables

**Figure 1 biomolecules-13-01516-f001:**
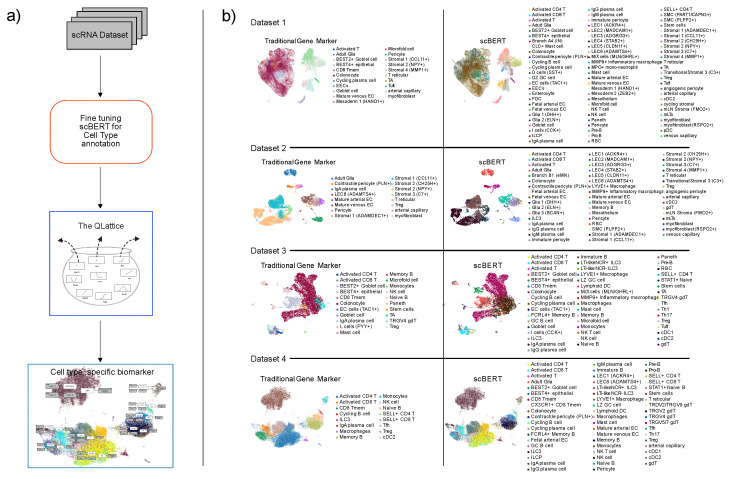
(**a**) illustrates the sequential steps employed in the workflow for identifying biomarkers specific to each cell type. (**b**) illustrates the refined granularity in cell type annotations resulting from the fine-tuning of scBERT in comparison to the conventional gene marker method across the four datasets. The annotations produced by scBERT not only include the same cell types identified by the original gene markers, but also introduce several additional ones, effectively addressing the core challenge of achieving consistent cell type annotations. The visuals on the left depict cell type labeling based on traditional gene markers, while those on the right reveal the augmented annotations obtained through scBERT fine-tuning.

**Figure 2 biomolecules-13-01516-f002:**
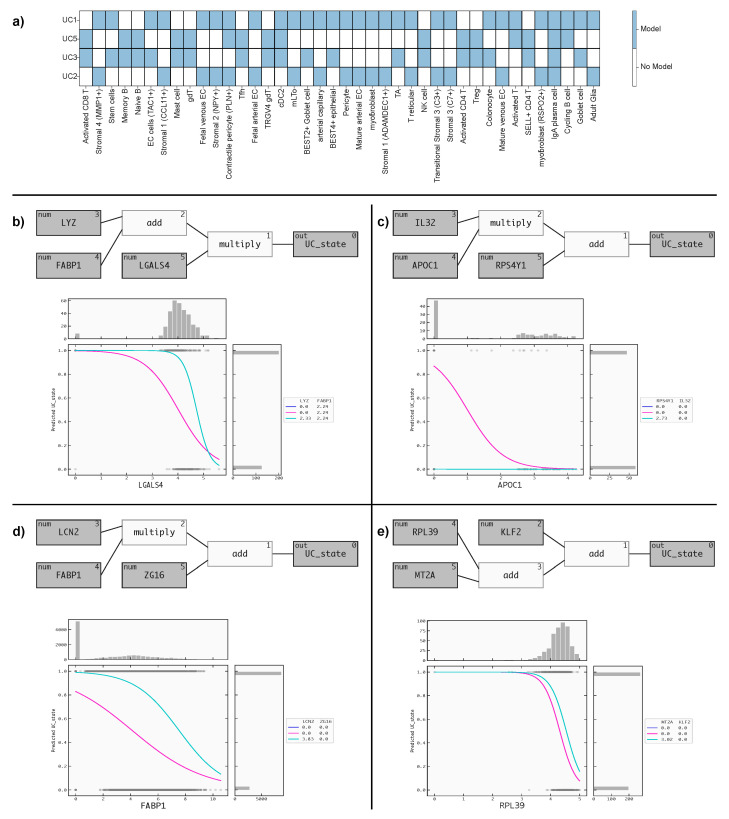
(**a**) illustrates the presence of various cell types within these datasets. (**b**–**e**) shows the QLattice models and response plots that serve to illustrate how the models have incorporated the disease state information.

**Table 1 biomolecules-13-01516-t001:** Summary of the performance of the best models across cell types and datasets. This table provides a comprehensive overview of our model’s performance across diverse cell types, employing various features and evaluating through PR AUC. The “Dominant Features” column highlights the gene that consistently emerges as the most-frequent among the 10 models generated by the QLattice. The “Score” column in the table represents the transferability-corrected performance score explained in detail in the methods section. Furthermore, the “UC Fraction” column delineates the proportion of cell types corresponding to disease samples within both the training and test datasets.

Training ID	Test ID	Cell Type	Dominant Features	PR AUC	Score	Count	UC Fraction
D1	D2	Arterial Capillary	RPL39, PLAT	0.95/0.94	0.94	1814/559	28%/22%
D1	D3	Goblet Cell	LYZ, LGALS4	0.94/0.92	0.91	1714/325	26%/38%
D1	D3	Intestinal Stem	B3GNT7, FABP1	0.95/0.89	0.86	14,930/2821	35%/36%
D1	D3	BEST2+ Goblet Cell	LYZ, FABP1	0.96/0.88	0.84	11,444/1436	24%/42%
D2	D1	Myofibroblast	APOC1, RPS4Y1	0.94/0.92	0.91	105/1189	45%/16%
D3	D1	BEST2+ Goblet Cell	FABP1, LCN2	0.92/0.94	0.91	1436/11,444	42%/24%
D3	D1	Intestinal Stem	FABP1, LCN2	0.93/0.91	0.90	2821/14,930	36%/35%
D3	D1	Stem Cell	LCN2, DDIT4	0.92/0.88	0.86	102/123	19%/21%
D4	D3	Activated CD8 T	KLF2, RPL39	0.82/0.84	0.81	3212/460	41%/43%

## Data Availability

**Dataset 1** “*Intra- and Inter-cellular Rewiring of the Human Colon during Ulcerative Colitis*”. https://data.humancellatlas.org/explore/projects/cd61771b-661a-4e19-b269-6e5d95350de6 (accessed on 1 April 2023) **Dataset 2** “*Structural remodeling of the human colonic mesenchyme in inflammatory bowel disease*”. https://www.ncbi.nlm.nih.gov/geo/query/acc.cgi?acc=GSE114374 (accessed on 1 April 2023) **Dataset 3** “*Colonic epithelial cell diversity in health and inflammatory bowel disease*”. https://www.ncbi.nlm.nih.gov/geo/query/acc.cgi?acc=GSE116222 (accessed on 1 April 2023) **Dataset 4** “*Heterogeneity and clonal relationships of adaptive immune cells in ulcerative colitis revealed by single-cell analyses*”. https://www.ncbi.nlm.nih.gov/geo/query/acc.cgi?acc=GSE125527 (accessed on 1 April 2023) **A codebase is available on Github** https://github.com/Jotels/LLM_xAI_Biomarkers (accessed on 31 July 2023)

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
