# Peer review of "Using LLMs and Explainable ML to Analyze Biomarkers at Single-Cell Level for Improved Understanding of Diseases"

_biomolecules, 2023, doi:10.3390/biom13101516_

Round 1

Reviewer 1 Report

In the manuscript biomolecules-2601544 entitled “Using LLM Models and Explainable ML to Analyse Biomarkers at Single Cell Level for Improved Understanding of Diseases”, authors used scBERT to establish consistent cell-type annotations across multiple scRNA-seq datasets, and employed a symbolic regression algorithm to pinpoint highly relevant yet minimally redundant models and features.

1.     In line 53,54 etc, many citations seemed to be missed, as “?” labels.

2.     In paragraph in line 109, authors stated “four diverse datasets on ulcerative colitis (UC), originating from various tissues”. However, there was not introduction on Dataset4, and there is a need to introduce the various tissues and their contributions in such study.

3.     In Figure 1, “Figure 1b shows the enhanced level of detail regarding cell types achieved through the fine tuning of scBERT compared to the conventional gene marker method across the four datasets.” But, this conclusion is not solid; it is required to provide quantitative comparison for the improvement on cell type annotation by scBERT.

4.     Authors analyzed four datasets separately, it is necessary to investigate all data in an integrative manner, because current analysis will produce non-consistent results. For example, as shown in Table 1, for Intestinal Stem, D1->D3 and D3->D1 will obtain different marker features. Similar difference was also observed on BEST2+ Goblet Cell. Thus, more consistent or robust analysis and results are required.

5.     A same question is that authors would obtain different models on different datasets, however, a unified model for the whole gut cell atlas should be provided.

Minor editing of English language required

Author Response

In line 53,54 etc, many citations seemed to be missed, as “?” labels. 

We greatly appreciate your observation regarding the issue and we corrected the errors.

In paragraph in line 109, authors stated “four diverse datasets on ulcerative colitis (UC), originating from various tissues”. However, there was not introduction on Dataset4, and there is a need to introduce the various tissues and their contributions in such study.

We appreciate the reviewer's attention to detail and their feedback regarding the introduction of Dataset 4 and the clarification of the various tissues used in our study.  We would like to clarify that a detailed description of the datasets, including Dataset 4, is provided in the dataset section of the manuscript. In this section, we have elaborated on the origins and characteristics of all four datasets related to ulcerative colitis (UC). This includes information on the tissues from which the data were derived.

In Figure 1, “Figure 1b shows the enhanced level of detail regarding cell types achieved through the fine tuning of scBERT compared to the conventional gene marker method across the four datasets.” But, this conclusion is not solid; it is required to provide quantitative comparison for the improvement on cell type annotation by scBERT.

We appreciate the reviewer's feedback and concerns regarding the need for a quantitative comparison to assess the improvement on cell type annotation achieved through the fine-tuning of scBERT. We apologize for any lack of clarity in text and we tried to make it clearer.

Our primary objective in Figure 1 was to visually demonstrate the enhanced granularity of cell type annotation made possible by fine-tuning scBERT in comparison to conventional gene marker methods. It is crucial to emphasize the significance of this aspect, especially when we take into account that the cell type annotations generated by scBERT encompass not only the same cell types as identified by the original gene markers but also numerous additional ones, thus addressing the challenge of achieving consistent cell type annotations. However, we fully acknowledge the importance of substantiating this claim more robustly through the inclusion of quantitative metrics. We have made efforts to address this by elaborating on our approach in the scBERT section of the manuscript.

Additionally, we would like to clarify our rationale for using scBERT in this study. Our intention was not to benchmark scBERT against existing methods, as the original authors of scBERT and the underlying framework have previously conducted comprehensive benchmarking. 

Our goal was to demonstrate the versatility and customizability of our approach by incorporating scBERT into the analysis pipeline. By doing so, we aimed to show how this advanced tool can be integrated into single-cell analysis workflows to enhance data granularity. Furthermore, it's important to note that our study's focus was on providing a general and adaptable methodology that can be adjusted as new tools and improvements emerge. We aimed to showcase the potential benefits of using scBERT or other LLM based-methods and provide a foundation for researchers to explore and customize their analyses based on evolving technologies and techniques.

Authors analyzed four datasets separately, it is necessary to investigate all data in an integrative manner, because current analysis will produce non-consistent results. For example, as shown in Table 1, for Intestinal Stem, D1->D3 and D3->D1 will obtain different marker features. Similar difference was also observed on BEST2+ Goblet Cell. Thus, more consistent or robust analysis and results are required.

We appreciate your comment regarding the necessity of investigating all four datasets in an integrative manner to ensure consistent and robust results. Our study's scope, influenced by the limited amount of patient data available, aimed to provide a robust outcome by leveraging the existing datasets effectively.

To ensure the robust evaluation of our models, we opted to train and analyze them independently using the four datasets. Our approach involved running the QLattice on a single dataset, from which we generated a set of models. Subsequently, these models were tested on the remaining three datasets. To maintain reproducibility, we implemented a fixed random seed for the QLattice. All 40 generated models, comprising 10 models per fold across the four datasets, are comprehensively presented in the supplementary tables.

It's important to clarify that our analysis's overarching goal was to identify gene expression signatures capable of distinguishing disease states and transferable across different datasets. To quantitatively measure this, we employed a metric that accounts for both absolute performance and transferability. Specifically, we utilized the average PR AUC (Precision-Recall Area Under the Curve) between the training and test datasets. We believe that this approach provides a holistic assessment, considering the performance similarity on both training and test datasets, thus ensuring that our models capture meaningful and consistent patterns despite the inherent data variability.

A same question is that authors would obtain different models on different datasets, however, a unified model for the whole gut cell atlas should be provided.

We appreciate your question regarding the generation of different models for various datasets. It's important to clarify that the primary objective of our paper is not to create a unified model for the entire gut cell atlas but, rather, to showcase the versatility of symbolic regression methods like the QLattice in unlocking multiple insights from diverse datasets and the high rate of generalizations and transferability due to the simple models. Our approach is intentionally designed to generate multiple models, each offering unique mathematical equations that reveal specific data patterns. This approach aligns with our goal of leveraging existing data and information from the gut cell atlas to uncover a range of cell type-specific biomarkers and associations. Additionally, the variation in models across datasets reflects the inherent heterogeneity and complexity of biological systems. We recognize that a unified model may not capture these nuances effectively, given the limited data available and our focus on extracting insights rather than creating a new gut cell atlas. 

We marked in red the change in the text to address some of these points in the text

Reviewer 2 Report

In the following study, Elsborg et al applies a symbolic explainable ML method to extract gene signatures that predict disease state. Furthermore they use a fine tuned scBERT for cell type annotation. The main strength of this paper is that it highlights how symbolic regression might be useful.

I see the use of scBERT more as a routine “supplemental” method needed for cell type annotation, although they go into detail in assessing transferability. I had personally downplayed this part, but it is fine as is

The paper has the style of a vignette, showing how QLattice can be used, which is not something I have come across before. I dont see any papers applying it to single cell data. The generated biological insight is limited but I don’t think this is an issue for the paper

The presentation is fine

The code is very well documented, which is crucial for this type of paper. However, Github links to https://drive.google.com/drive/folders/1jyl0esSY8VYMQ1Ke7P-E-VMidR022awM?usp=drive_link which does not strike me as very stable. The data should be deposited at zenodo, figshare, or similar

The study has one enormous weakness, which either has to be discussed, or better, analyzed in more detail. This is that QLattice generates /one/ model, which need not be the most useful model. The inclusion of RPL39 is a good example, because this may not be the most informatice gene. I suspect many other RPL* genes could have been included as well. The study could be greatly boosted if an analysis was made on alternative models. I.e. if the user does not like a certain gene, and it is removed, would a completely different model be picked, or could just that gene be replaced? In terms of interpretability, it is crucial that the model includes genes that we know pretty well. In single cell analysis, it is pretty common that the top ranked marker genes are not such genes

It is crucial that the paper also makes a note that the genes included in the models need /not/ be causal, even if it may happen in the lucky case

In summary, the paper can be accepted as-is with above disclaimers added, but an elaboration on alternative models would further lift the study

Author Response

The code is very well documented, which is crucial for this type of paper. However, Github links to https://drive.google.com/drive/folders/1jyl0esSY8VYMQ1Ke7P-E-VMidR022awM?usp=drive_link which does not strike me as very stable. The data should be deposited at zenodo, figshare, or similar

Thank you for pointing out this issue.  We converted the drive repository to a zenodo one that can be found here 10.5281/zenodo.8371114

The study has one enormous weakness, which either has to be discussed, or better, analyzed in more detail. This is that QLattice generates /one/ model, which need not be the most useful model. The inclusion of RPL39 is a good example, because this may not be the most informatice gene. I suspect many other RPL* genes could have been included as well. The study could be greatly boosted if an analysis was made on alternative models. I.e. if the user does not like a certain gene, and it is removed, would a completely different model be picked, or could just that gene be replaced? In terms of interpretability, it is crucial that the model includes genes that we know pretty well. In single cell analysis, it is pretty common that the top ranked marker genes are not such genes.

We appreciate the thoughtful feedback and acknowledge the concern about the study's reliance on a single model generated by QLattice. Indeed, this is an important aspect to consider in our methodology.

In our study, we chose to present the top-performing model, focusing on performance, simplicity, and transferability as our primary criteria. This decision aimed to provide a clear and practical approach for researchers seeking insights from single-cell data, ensuring that the results are readily interpretable and usable.

However, it's essential to emphasize that our framework does allow for the generation of multiple models. We have included all the generated models in the supplementary table to provide transparency and enable further investigation by researchers interested in alternative approaches or different gene selections.

Regarding the inclusion of specific genes, such as RPL39, we understand the concern about the potential inclusion of genes that may not be the most informative. Our study does not preclude the possibility of exploring alternative gene selections or removing genes that are of less interest to users. Researchers can certainly use our framework to tailor the analysis to their specific interests and domain knowledge.

Ultimately, the scope of our paper was to demonstrate an alternative and agnostic method for leveraging single-cell information to gain insights into diseases. We encourage researchers to explore the flexibility of our framework and consider different gene selections or alternative models to address specific research questions or hypotheses of interest.

In summary, while we focused on presenting the top-performing model for practicality, the supplementary table contains all generated models, and our framework allows for customization and exploration, thereby offering flexibility and adaptability to suit different research needs.

We marked in red the change in the text

It is crucial that the paper also makes a note that the genes included in the models need /not/ be causal, even if it may happen in the lucky case.

We greatly appreciate your observation regarding the issue of causal relationships in our study. It's important to clarify that our research was intentionally designed within a specific scope, which did not involve establishing causal relationships. Instead, our primary objective was to leverage single-cell information and use an alternative and agnostic approach to uncover potential associations and insights into diseases.

Causality is a complex and multifaceted concept in biomedical research. Establishing causal relationships typically involves integrating various types of data, including genetic, among others. Such an endeavor would require a broader and more comprehensive research framework than the one presented in our study.

In recognition of this limitation, we have taken steps to make it clear in the text that our study's focus was on correlation and association rather than causation. We understand that causation is a critical aspect of understanding disease mechanisms, but our study serves as a foundational step toward identifying candidate genes and relationships that warrant further investigation.

We marked in red our changes in the text.

Round 2

Reviewer 1 Report

Authors have responded to all my concerned questions, and made a good revision.

There is a still one issue:

Authors claimed that they focused on cellular heterogeneity in various biological contexts in the revision and responses. Thus, it is better to discuss the identification of biological variation and technical variation in different models, so as to recognize meaningful biological findings by multiple computational models rather than a unified model.

NA

Author Response

We once again thank the reviewer for an insightful suggestion. The reviewer suggestion prompted us to delve deeper into exploring the potential role of biological variation in elucidating the distinctions observed in various models and features across diverse cell types. Consequently, we chose to investigate an additional model, focusing on a unique dominant gene—specifically, PLAT.  In our exploration as described in the new paragraph 3.4, we highlighted instances where PLAT gene expression in the arterial capillary model can be linked to well-established pathological mechanisms in ulcerative colitis. This observation led us to recognize that PLAT expression holds particular significance in cells from blood vessels(which is also the case in our study), which thus indeed shows that biological variation is meaningful in the context of predictive models on data from different cell types"

Reviewer 2 Report

All my issues have been addressed. I find the language in the new textual edits somewhat overcomplicated, but it hopefully gets the point through

Author Response

We would like to thank the reviewer to provide meaningful observations and changes.